# Enzyme immunoassays (EIA) for serodiagnosis of human leptospirosis: specific IgG3/IgG1 isotyping may further inform diagnosis of acute disease

Elsa Fortes-Gabriel[1,2], Mariana Soares Guedes[2¤a], Advait Shetty[3], Charles Klazer Gomes[2¤b], Teresa Carreira[4], Maria Luísa Vieira[4], Lisa Esteves[5], Luísa Mota-Vieira[5,6], Maria Gomes-Solecki[2,3]*

1 Instituto Superior Técnico Militar—Estado Maior General das Forças Armadas Angolanas, Luanda, Angola, 2 Immuno Technologies Inc, Memphis, Tennessee, United States of America, 3 Department of Pharmaceutical Sciences—University of Tennessee Health Science Center, Memphis, Tennessee, United States of America, 4 Instituto de Higiene e Medicina Tropical—Universidade Nova de Lisboa, Lisbon, Portugal, 5 Molecular Genetics and Pathlogy Unit—Hospital do Divino Espírito Santo de Ponta Delgada, São Miguel Island—Azores, Portugal, 6 Azores Genetics Research Group—Instituto Gulbenkian de Ciência, Oeiras, Portugal

¤a Current address: Department of Microbiology and Molecular Medicine, University of Geneva, Switzerland
¤b Current address: Dynavax, Emeryville, California, United States of America
* mgomesso@uthsc.edu

**Data Availability Statement:** All relevant data are within the manuscript and its Supporting Information files.

## Abstract

The laborious microscopic agglutination test (MAT) is the gold standard serologic test for laboratory diagnosis of leptospirosis. We developed EIA based serologic assays using recombinant proteins (rLigA, rLigB, rLipL32) and whole-cell extracts from eight *Leptospira* serovars as antigen and assessed the diagnostic performance of the new assay within each class, against MAT positive (MAT+) human sera panels from Portugal/PT (n = 143) and Angola/AO (n = 100). We found that a combination of recombinant proteins rLigA, rLigB and rLipL32 correctly identified antigen-specific IgG from patients with clinical and laboratory confirmed leptospirosis (MAT+) with 92% sensitivity and ~ 97% specificity (AUC 0.974) in serum from the provinces of Luanda (LDA) and Huambo (HBO) in Angola. A combination of whole cell extracts of *L. interrogans* sv Copenhageni (LiC), *L. kirschneri* Mozdok (LkM), *L. borgpetersenii* Arborea (LbA) and *L. biflexa* Patoc (LbP) accurately identified patients with clinical and laboratory confirmed leptospirosis (MAT+) with 100% sensitivity and ~ 98% specificity for all provinces of Angola and Portugal (AUC: 0.997 for AO/LDA/HBO, 1.000 for AO/HLA, 0.999 for PT/AZ and 1.000 for PT/LIS). Interestingly, we found that MAT+ IgG+ serum from Angola had a significantly higher presence of IgD and that IgG3/IgG1 isotypes were significantly increased in the MAT+ IgG+ serum from Portugal. Given that IgM/IgD class and IgG3/IgG1 specific isotypes are produced in the earliest course of infection, immunoglobulin G isotyping may be used to inform diagnosis of acute leptospirosis. The speed, ease of use and accuracy of EIA tests make them excellent alternatives to the laborious and expensive MAT for screening acute infection in areas where circulating serovars of pathogenic *Leptospira* are well defined.

**Funding:** This work was supported by the Public Health Service awards AI155211 (MGS), AI142129 (MGS) and AI136551 (MGS) from the National Institute of Allergy and Infectious Diseases (NIAID) of the National Institutes of Health (NIH) of the United States of America. The funders had no role in study design, data collection and analysis, decision to publish, or preparation of the manuscript.

**Competing interests:** I have read the journal's policy and the authors of this manuscript have the following competing interests: MG-S (grants from federal agencies and employment). All other authors declare no conflicts.

## Author summary

Leptospirosis is a potential fatal zoonotic disease with worldwide distribution caused by a bacterium found in contaminated sources of water and soil. Diagnosis of leptospirosis is tentatively based on evaluation of fever and myalgia in patients presenting at the hospital in areas of endemicity, and it is rarely confirmed in most parts of the world due to lack of affordable diagnostic tests. We developed a highly sensitive and specific serologic test for leptospirosis using ubiquitous enzyme immunoassay technology. In areas where circulating serovars of pathogenic *Leptospira* are well defined, versatile enzyme immunoassays can be adapted and operated at a fraction of the cost of the cumbersome gold standard Microscopic Agglutination Test and should be explored as accurate serodiagnostic tools for leptospirosis. Addition of specific immunoglobulin G isotyping (IgG3/IgG1) can further inform diagnosis of acute leptospirosis.

## Introduction

Leptospirosis is a neglected emerging zoonotic disease with worldwide distribution that affects essentially all vertebrates, mostly in resource-poor and developing countries [1]. It affects vulnerable populations such as rural subsistence farmers and urban slum dwellers. Urban epidemics are reported mostly in cities of developing countries and will likely increase as the world's slum population doubles to 2 billion by 2030 [1]. In developed countries, unexpected deadly outbreaks have been reported in New York City in 2017 [2] and 2021 [3]. However, outbreaks of leptospirosis are expected in grain-growing rainy regions of Australia when reservoir host populations of house mice skyrocket [4]. Sub-Saharan African countries lack notification surveillance systems for leptospirosis and in most cases the laboratory diagnosis is not done [5]. The estimated prevalence from countries that report the disease is high (75–102 per 100,000 population in Tanzania) compared with occidental countries [6,7]. Laboratory diagnosis of leptospirosis is not established in many Sub-Saharan countries despite the need for differential diagnosis with malaria, dengue, yellow fever and other common febrile illnesses [8,9], as is the case in Angola [10].

Human leptospirosis ranges in severity from a mild, self-limited febrile illness to a fulminant life-threatening disease [11]. A review of published cases estimated that leptospirosis causes ~1 million cases a year [1], resulting in ~5–10% death rate [1,12]. A number of organs is involved, reflecting the systemic nature of the infection. As a result, the symptoms of leptospirosis are frequently mistaken for other causes of acute febrile syndrome such as dengue, hepatitis [11], and malaria, depending on the overlap of endemic geographic areas. Unlike other spirochetal diseases that are characterized by signs and symptoms that aid in clinical diagnosis (ex. the bull's eye erythema migrans in Lyme disease, and the chancre in primary syphilis), diagnosis of leptospirosis is tentatively based on evaluation of fever and myalgia in patients presenting at the hospital in areas of endemicity, and it is rarely confirmed in most parts of the world due to lack of affordable diagnostic assays.

The microscopic agglutination test (MAT) is the current gold standard serologic test for laboratory diagnosis of leptospirosis. However, it cannot be used for immediate case identification because it is insensitive in early infection [11] and it can only be performed in specialized laboratories with highly trained personnel and specific conditions such as reference collections of live *Leptospira* serovars, darkfield microscopy, among others. Thus, it is rarely performed by clinical diagnostic laboratories [13]. Less cumbersome, accurate, affordable, and accessible

serologic tests are needed to better inform leptospirosis management [14]. We developed Enzyme ImmunoAssays (EIA) using either recombinant proteins or *Leptospira* whole-cell extracts as antigen and assessed the diagnostic performance of the new assay using MAT+ sera panels from Portugal and Angola.

## Methods

### Ethics statements

**Humans.** The present study followed international ethical guidelines and was evaluated and approved for use of human serum by the Health Ethics Committee of the Hospital do Divino Espírito Santo de Ponta Delgada, Azores, EPER (HDES/CES/159/2009), by the National Data Protection Commission authorization granted according to WHO guidelines to the reference laboratory for human leptospirosis diagnosis in Portugal (Leptospirosis and Lyme Borreliosis Laboratory—LLB-Lab) at the Instituto de Higiene e Medicina Tropical/ IHMT (no.55/2000), and by the scientific board of the National Ethical Committee of Health of Angola (Ref. MINSA/CES/04/2011). The involvement of human subjects in this study falls under exemption 4 as outlined in the US HHS regulations (45 CFR Part 46). A total of 286 de-identified human serum samples were used. Serum was collected between 2009–2015 and was kept at -20˚C without glycerol until use in 2018/2019. IRB was approved under FWA00021769 by Integreview/Advarra Ethical Review Board #2.

**Animals.** Eight-week-old female C3H/HeJ mice (Charles River, Boston) were acclimatized for one week in the pathogen-free environment in the UTHSC LACU before infection with *L. interrogans* FioCruz. Experimental animals were used per the guide for Care and Use of Laboratory Animals of the National Institutes of Health under approved protocol (19–0062) of the University of Tennessee Health Science Center Institutional Animal Care and Use Committee.

### Geographic areas

Human serum samples from an African country (Southern Hemisphere, Angola—AO, provinces of Luanda—LDA, Huambo—HBO, and Huila—HLA) and from a Western European country (Northern Hemisphere, Portugal–PT, mostly from the Azores islands–AZ, and from the mainland—LIS) were used in this study. These different geographic areas are quite distinct in economic development and climate.

### Serum panels

**AO (LDA/HBO).** A panel of 50 human serum samples from individuals with fever and laboratory-confirmed leptospirosis (MAT+ titer > 1:100) collected between 2010 and 2012 in the provinces of Luanda and Huambo in Angola.

**AO (HLA).** A panel of 50 human serum samples from individuals with fever and laboratory-confirmed leptospirosis (MAT+ titer > 1:100) collected between 2012 and 2013 in the province of Huila in Angola.

**PT (AZ).** A panel of 43 human serum samples from individuals with clinical symptoms (fever, myalgia) and laboratory-confirmed leptospirosis (MAT+ titer > 1:100) collected between 2009 and 2015 at the HDES, São Miguel, Azores, Portugal.

**PT (LIS).** A panel of 100 human serum samples from individuals with clinical symptoms (fever, myalgia) and laboratory-confirmed leptospirosis (MAT+ titer > 1:100) collected at the LLB-Lab/IHMT, Lisbon, Portugal. This panel contains samples collected between 2009 and 2015 in several regions of mainland Portugal (Coimbra, Lisbon, and Évora), and from the Terceira Island in the Azores.

**Healthy.**  A panel of 43 human serum samples from healthy individuals from a non-endemic area for leptospirosis from the US (Florida) acquired in 2018.

*Murine serum*: a panel of 12 serum samples from 6 mice infected intraperitoneally with $10^7$ *L. interrogans* FioCruz and from 6 non-infected controls.

## Antigens

We used a commercial service (GenScript Biotech, Piscataway, NJ) to clone ten *L. interrogans* genes (*flaB, flgC, flgJ, fliE, ligA_{7-13}, ligB_{1-6}, loa22, lemA, lipL32, tolC*) in pET plasmid vectors containing 6xHis-tags. We used the following eight serovars of *Leptospira* obtained from different sources: *L. interrogans* serovar (sv) Copenhageni L1-130 (human isolate from Brazil, a gift from Dr. D. Haake), *L. interrogans* Pomona sv Pomona (human isolate from Australia, ATCC 23478), *L. kirschneri* Pomona sv Mozdok 61H (human isolate from Brazil, a gift from Dr. O. Dellagostin), *L. kirschneri* sv Cynopteri 3522C (bat isolate from Indonesia, LLB-Lab/ IHMT collection), *L. borgpetersenii* sv Castellonis (wood mouse isolate from Spain, ATCC #23580), *L. borgpetersenii* sv Hardjo Type Bovis (bovine isolate, a gift from Dr. J. Nally), *L. borgpetersenii* Ballum sv Arborea (wood mouse isolate from Italy, LLB-Lab/IHMT collection), *L. biflexa* sv Patoc (soil isolate from France, ATCC #23582).

## Purification of recombinant proteins

Plasmids were used to transform *E. coli* BL21(DE3)pLysS competent cells (New England Biolabs, Ispwich, MA) containing vector specific selective antibiotic markers (Kan+ or Amp+) and the plates were incubated overnight at 37˚C; individual colonies cultured in TBY supplemented with antibiotic were induced with IPTG, cells were harvested by centrifugation, digested with BugBuster reagent (Millipore, Billerica, MA) in the presence of protease K inhibitor and the following recombinant proteins (rFlaB, rFlgC, rFlgJ, rFliE, rLigA_{7-13}, rLigB_{1-6}, rLoa22, rLemA, rLipL32, rTolC) were purified by affinity chromatography using HisPur Cobalt Resin (Thermo Fisher Scientific, Waltham, MA). Purity was checked by molecular weight size (KDa) on 10% denaturing polyacrylamide gel stained with Coomassie Blue and after electrotransfer to PVDF (Millipore, Billerica, MA) followed by western blot analysis using a mouse antigen-specific polyclonal antibody (1:100) and anti-mouse secondary antibody (1:1000) labeled with alkaline phosphatase. Examples of purified rLigA_{7-13}, rLigB_{1-6} and rLipL32 are provided in **S1 Fig**.

## Production of whole-cell extract antigens

*Leptospira* spp previously maintained in EMJH liquid medium were sub-cultured in EMJH semi-solid medium supplemented with 10% Difco *Leptospira* enrichment (Thermo Fisher Scientific, Waltham, MA), 1% of 5-Fluorouracil, and 2% inactivated rabbit serum (56˚C, 1h) and grown for 15–30 days at 29˚C shaking (100 rpm) in the dark. The individual cultures successfully recovered were grown in 25mL EMJH liquid for 5–10 days (to $10^8$ Cells/mL) and then centrifuged at 9,000 rpm for 15 min at room temperature (RT); the supernatant was discarded leaving an aliquot of 1mL which was centrifuged at 20,000 rpm (3 min) and the supernatant was discarded. The pellet was incubated with 1mL of BugBuster solution (protein extraction reagent containing nuclease and lysozyme) at RT in a shaking incubator (100 rpm) for 20 min, homogenized by vortex, and stocks were saved at -20˚C.

## Serologic testing on Enzyme ImmunoAssay (EIA)

EIA was performed according to a comprehensive published method (Protocol 3.3.1. in CPM [15]) with the following modifications (a step-by-step protocol is provided in **S1 Text**). Briefly,

purified recombinant proteins quantified by the Lowry protein assay kit (Thermo Fisher Scientific, Waltham, MA) or whole-cell extract of *Leptospira* diluted in 1X sodium carbonate coating buffer was used to coat Nunc MaxiSorp flat-bottom EIA plates (eBioscience, San Diego, CA) at 0.5–1 µg/ml (protein) or $10^5$-$10^8$ cells/well (*Leptospira* extract) overnight at 4°C. The following day the plates were washed with 1XPBS, blocked with 1% BSA for 2h, washed again, and incubated with human or murine serum diluted at 1:50 or 1:100. Goat anti-mouse or goat anti-human IgG conjugated to HRP diluted at 1:10000 (Jackson ImmunoResearch, West Grove, PA) was used as the secondary antibody. The $OD_{450}$ was read on a SpectraMax Plus EIA reader (Molecular Devices). For determination of Ig class and IgG isotypes the secondary antibody conjugated to HRP was diluted as follows: anti-human -IgG1, -IgG2, -IgG3, -IgG4 (1:1000, Invitrogen), anti-human IgA (1:8000, Southern Biotech), anti-human IgD (1:1200, Southern Biotech) and IgM (1:10,000, Jackson ImmunoResearch Laboratories, Inc.). The EIA cutoff was set at three standard deviations above the average $OD_{450}$ of all healthy control samples.

## Statistical analysis

For analysis of diagnostic performance, the OD cutoff for each antigen was determined for each leptospirosis panel versus healthy control by constructing a receiver-operating characteristic (ROC) analysis with area under the curve (AUC), and selecting the OD cutoff value which resulted in the maximum sensitivity given a minimum of 97% specificity. Furthermore, Ordinary One-Way ANOVA was used to assess the significance of differences between each of the leptospirosis panels and the healthy control. For the combined algorithm, we chose the recombinant protein or whole cell extract with the highest sensitivity and then included the negative samples that tested positive against any other biomarker candidate (a positive in 1 of the 3 or 4 biomarkers was considered positive). For analysis of differences between two IgG+ panels for Ig class and IgG isotyping the Welch's t test was used.

## Results

### Screening recombinant protein candidates for specificity and potential sensitivity

To evaluate EIA specificity of the 10 candidate biomarkers we tested purified recombinant proteins using *L. interrogans* whole cell extract (WCE) as control, against serum (1:50) from healthy individuals from an area (FL/US) that is not endemic for leptospirosis (**Fig 1**). We found that seven recombinant proteins (rFlaB, rFlgC, rLigA$_{7-13}$, rLigB$_{1-6}$, rLemA, rLipL32 and rTolC) were more specific (less cross-reactive) to the healthy human serum than the WCE control. Of these seven, FlaB, LigA, LigB and LipL32 were shown to be reactive with serum from leptospirosis patients [16], and three (LigA, LigB, LipL32) had 95–99% homology between all pathogenic *Leptospira* serovars [17]. To confirm the potential sensitivity of the three leads in comparison with other candidates, we tested purified rLigA, rLigB, rLipL32, rTolC and rFliE using serum from C3H-HeJ mice infected with *L. interrogans* sv Copenhageni FioCruz. We confirmed that unlike TolC and FliE, the three recombinant proteins LigA, LigB and LipL32 were highly immunogenic in mice (**S2 Fig**).

### Diagnostic performance of recombinant proteins and *Leptospira* whole-cell extracts against human leptospirosis serum panels from two distinct geographic regions

Four MAT+ leptospirosis serum panels from two countries, representing Northern and Southern Hemispheres geographic regions, were used to evaluate the diagnostic potential of each

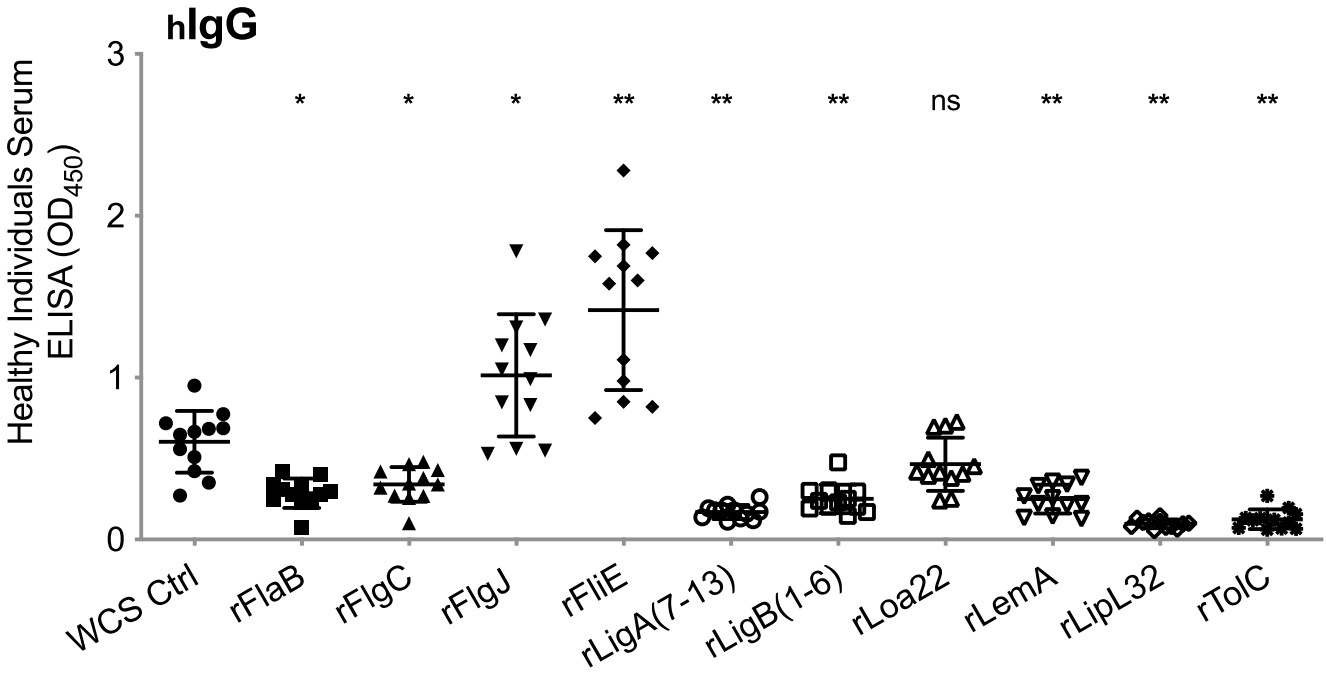

**Fig 1. Specificity analysis of candidate serodiagnostic biomarkers.** Human IgG specific to purified *L. interrogans* recombinant proteins compared to *L. interrogans* whole cell extract (WCE) using a serum panel from healthy individuals (n = 12 samples, diluted 1:50) from a non-endemic area for leptospirosis (FL, US). Statistics by Mann-Whitney test (unpaired, exact): comparison between each recombinant protein and WCE, * p<0.005, ** p<0.0001, ns, not significant.

antigen (recombinant protein and whole-cell extract) by measuring their sensitivity and specificity. We tested the three lead recombinant proteins (rLigA, rLigB and rLipL32) selected after our initial specificity analysis against four distinct serum panels by EIA and found that if keeping a specificity >96%, each recombinant protein detected each serum panel with different sensitivities (**Fig 2, Table 1** and **S1 Data**). For Angola (AO), rLigA and rLipL32 performed better than rLigB: rLigA detected MAT+ serum with 88% sensitivity for LDA/HBO (AUC 0.958)

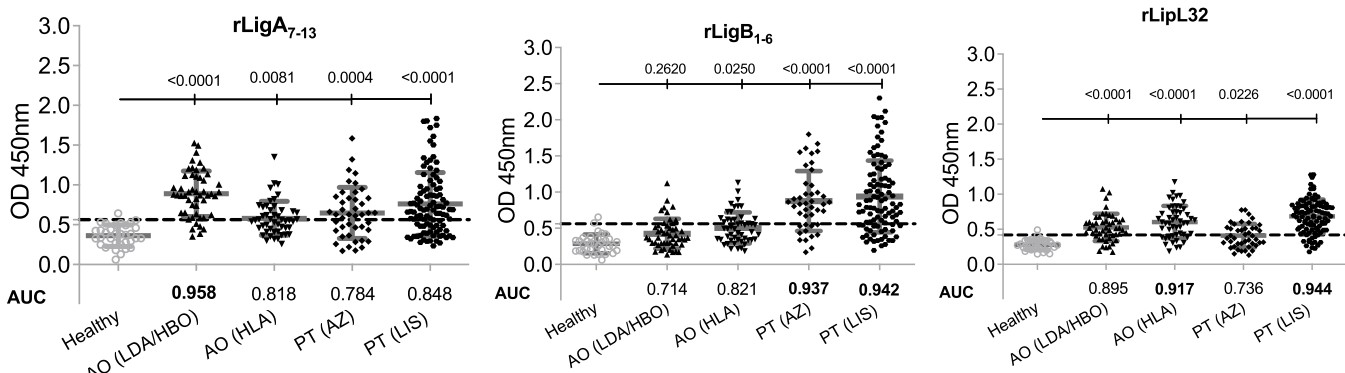

**Fig 2. Diagnostic performance of each *L. interrogans* recombinant protein.** Human IgG specific to rLigA, rLigB and rLipL32 in four MAT+ leptospirosis sera panels, AO (LDA/HBO) n = 49/50, AO (HLA) n = 49/50, PT (AZ) n = 43, PT (LIS) n = 100 and one control panel, Healthy n = 33/34. Statistics by Ordinary One-Way ANOVA between each leptospirosis panel and healthy control and by ROC Area Under the Curve (AUC).

**Table 1. Specificity and Sensitivity of Recombinant Proteins.**

| | SPECIFICITY | SENSITIVITY | | | |
|---|---|---|---|---|---|
| | **Healthy** | **Angola (LDA/HBO)** | **Angola (HLA)** | **Portugal (AZ)** | **Portugal (LIS)** |
| **rLigA** | *96.97%* | *88.00%* | *46.94%* | *58.14%* | *67.00%* |
| **rLigB** | *97.06%* | *34.69%* | *32.00%* | *72.09%* | *73.00%* |
| **rLipL32** | *97.06%* | *73.47%* | *76.00%* | *44.19%* | *87.00%* |
| **rLigA/rLigB/rLipL33** | *96.97%* | *92.00%* | *76.00%* | *74.42%* | *88.00%* |

LDA/HBO, Luanda/Huambo; HLA, Huila; AZ, Azores, LIS, Lisbon.

and rLipL32 detected MAT+ serum with 76% sensitivity for HLA (AUC 0.917). For Portugal (PT), rLigB and LipL32 fared much better than rLigA: rLigB detected MAT+ serum with 72% and 73% sensitivity for AZ and LIS (AUC 0.937 and 0.942, respectively) and rLipL32 detected MAT+ serum with 87% sensitivity for LIS (AUC 0.944).

We did a side-by-side analysis of each serum sample against each of the three recombinant antigens to assess if an improvement in sensitivity could be measured when screening the MAT+ samples against one, two or the three proteins. A sample that was positive in 1 of the 3 biomarkers was considered positive. We found that this combination dramatically increased the diagnostic performance of the assay (**Fig 3**, **Table 1** and **S1 Data**). While each recombinant protein detected the Angola AO (LDA/HBO) and AO (HLA) serum panels with sensitivities ranging from 35% to 88% and 32% to 76%, respectively, inclusion of positive samples to one of

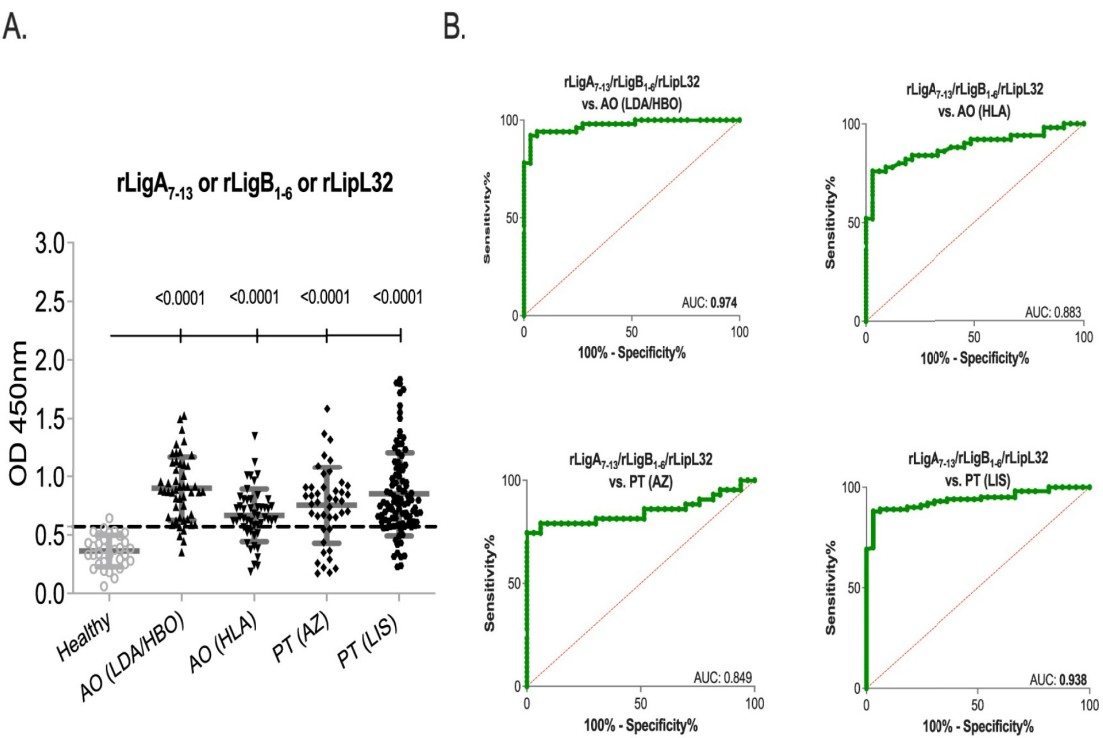

**Fig 3. Diagnostic performance of combined recombinant proteins.** Human IgG specific for rLigA or LigB or LipL32 in four MAT + leptospirosis sera panels, AO (LDA/HBO) n = 50, AO (HLA) n = 50, PT (AZ) n = 43, PT (LIS) n = 100 and one control panel, Healthy n = 33. A. Scatter plot of all sera panels and B. ROC curve of each sera panel. Statistics by Ordianry One-Way ANOVA and ROC AUC.

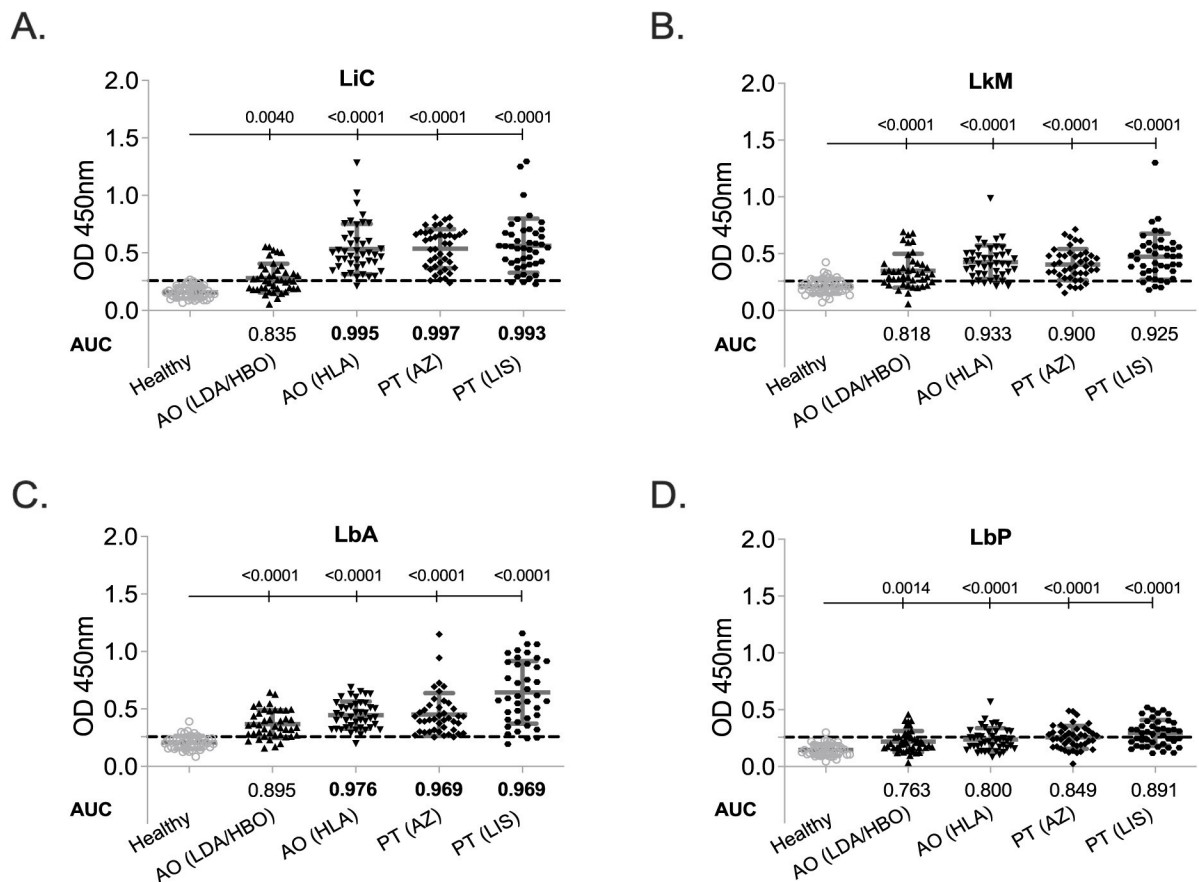

**Fig 4. Diagnostic performance of each *Leptospira* serovar extract.** Human IgG specific to *L. interrogans* Copenhageni FioCruz L1-130 (LiC), *L. kirschneri* Mozdok 61H (LkM), *L. borgpetersenii* Arborea (LbA) and *L. biflexa* Patoc 1 (LbP) in four MAT+ leptospirosis sera panels, AO (LDA/HBO) n = 43, AO (HLA) n = 43, PT (AZ) n = 42–43, PT (LIS) n = 40–43 and one control panel, Healthy n = 43. Statistics by Ordinary One-Way ANOVA between each leptospirosis panel and healthy control and by ROC Area Under the Curve (AUC).

the three biomarkers (LigA or LigB or LipL32) detected the same panels with 92% (AUC 0.974) and 76% sensitivity (AUC 0.883). The same trend was observed for the Portugal (PT) panels: single recombinant protein detection ranged from 44% to 72% (PT AZ) and 67% to 87% (PT LIS), whereas inclusion of positive samples to one of the three biomarkers (LigA or LigB or LipL32) resulted in 74% (AUC 0.849) and 88% (AUC 0.938) sensitivities for the same panels.

Next, we tested whole cells extracts from eight *Leptospira* serovars against the same serum panels by EIA and found that keeping a specificity >97%, each extract detected the four panels with different sensitivities (**Fig 4**, **Tables 2** and **S1** and **S1 Data**). For Angola, two whole-cell extracts performed well (sensitivity >90%) only against the sera panel from HLA: *L. interrogans* sv Copenhageni and *L. borgpetersenii* sv Arborea detected MAT+ serum with 97.67% and 93% sensitivity (AUC 0.995 and 0.976, respectively). For Portugal, *L. interrogans* sv Copenhageni was the best whole cell extract detecting MAT+ sera from AZ and LIS with 97.67% and 93% sensitivity (AUC 0.997 and 0.993, respectively).

We did a side-by-side analysis of each serum sample against each of the four *Leptospira* extracts to assess if an improvement in sensitivity could be measured when screening the MAT + samples against one, two, three or four whole cell extracts. A sample that was positive to 1 of the 4 biomarkers (LiC or LbA or LkM or LbP) was considered positive. We found that this

**Table 2. Specificity and Sensitivity of *Leptospira* Whole Cell Extracts.**

| | SPECIFICITY | SENSITIVITY | | | |
|---|---|---|---|---|---|
| | Healthy | Angola (LDA/HBO) | Angola (HLA) | Portugal (AZ) | Portugal (LIS) |
| *L. interrogans* Copenhagenii L1-130 | 97.73% | 53.49% | 97.67% | 97.67% | 93.02% |
| *L. kirschneri* Mozdok | 97.67% | 46.51% | 72.09% | 67.44% | 76.74% |
| *L. borgpetersenii* Arborea | 97.67% | 62.79% | 93.02% | 73.81% | 87.50% |
| *L. biflexa* Patoc1 | 97.67% | 37.21% | 48.84% | 62.79% | 67.44% |
| *LiC or LbA or LkM or LbP* | 97.67% | 100.00% | 100.00% | 100.00% | 100.00% |

LiC, *L. interrogans* Copenhageni; LbA, *L. borgpetersenii* Arborea, LkM, *L. kirschneri* Mozdoc; LbP, *L. biflexa* Patoc.

combination improved the diagnostic performance of the assay for all 4 sera panels tested (**Fig 5**, **Table 2** and **S1 Data**) with the largest gain in the AO (LDA/HBO) sensitivity which increased from 62.69% to 100% (AUC 0.997).

Lastly, we analyzed IgM, IgD and IgA class, and IgG isotypes between two MAT+, LiC IgG+ serum panels, one from Portugal (PT-AZ) and one from Angola (AO-HLA) (**Fig 6** and **S1 Data**). We found that MAT+ IgG+ serum from Angola had a significantly higher presence of IgD than MAT+ IgG+ serum from Portugal; and that IgG3/IgG1 isotypes were significantly increased in the MAT+ IgG+ serum from Portugal in contrast MAT+IgG+ serum from Angola. As expected, there were no significant differences in IgM or IgA between the two panels which were included based on IgG positivity to LiC.

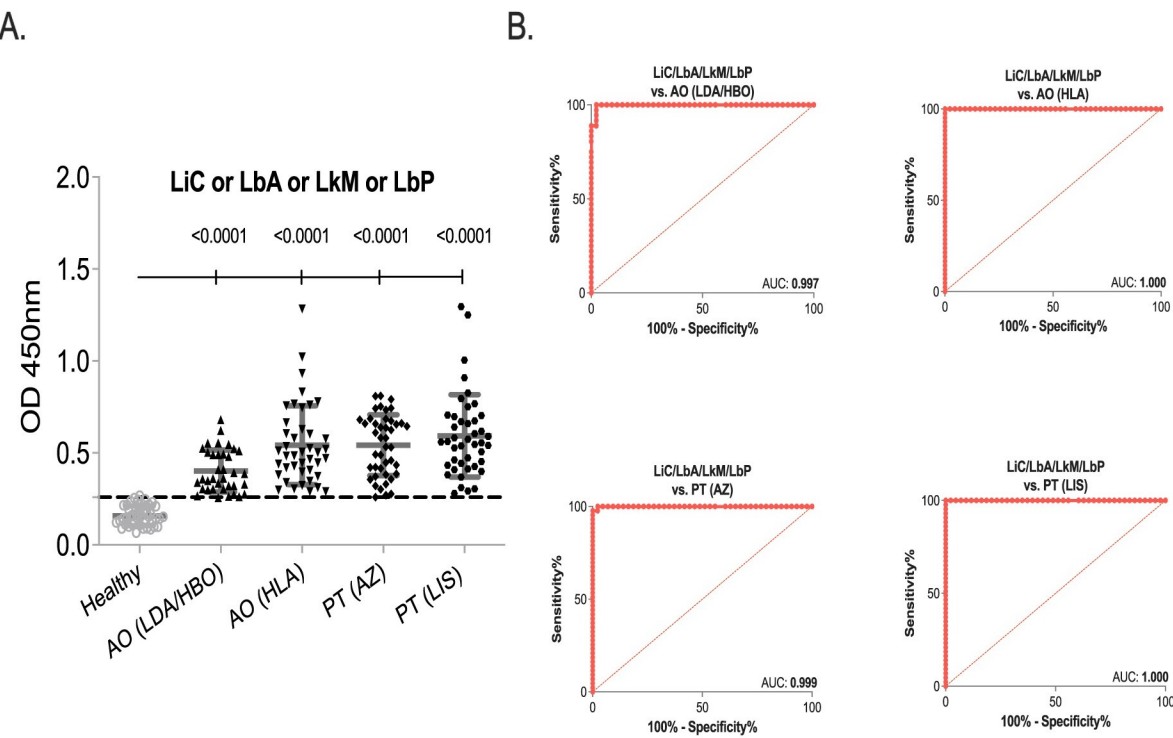

**Fig 5. Diagnostic performance of combined extracts of *Leptospira* serovars.** Human IgG specific to *L. interrogans* Copenhgeni FioCruz L1-130 (LiC), *L. borgpetersenii* Arborea (LbA), *L. kirschneri* Mozdoc 61H (LkM) and *L. biflexa* Patoc 1 (LbP) in four MAT+ leptospirosis sera panels, AO (LDA/HBO) n = 36, AO (HLA) n = 42, PT (AZ) n = 43, PT (LIS) n = 43 and one control panel, Healthy n = 43. A. Scatter plot of all sera panels and B. ROC curve of each sera panel. Statistics by Ordinary One-Way ANOVA and ROC AUC.

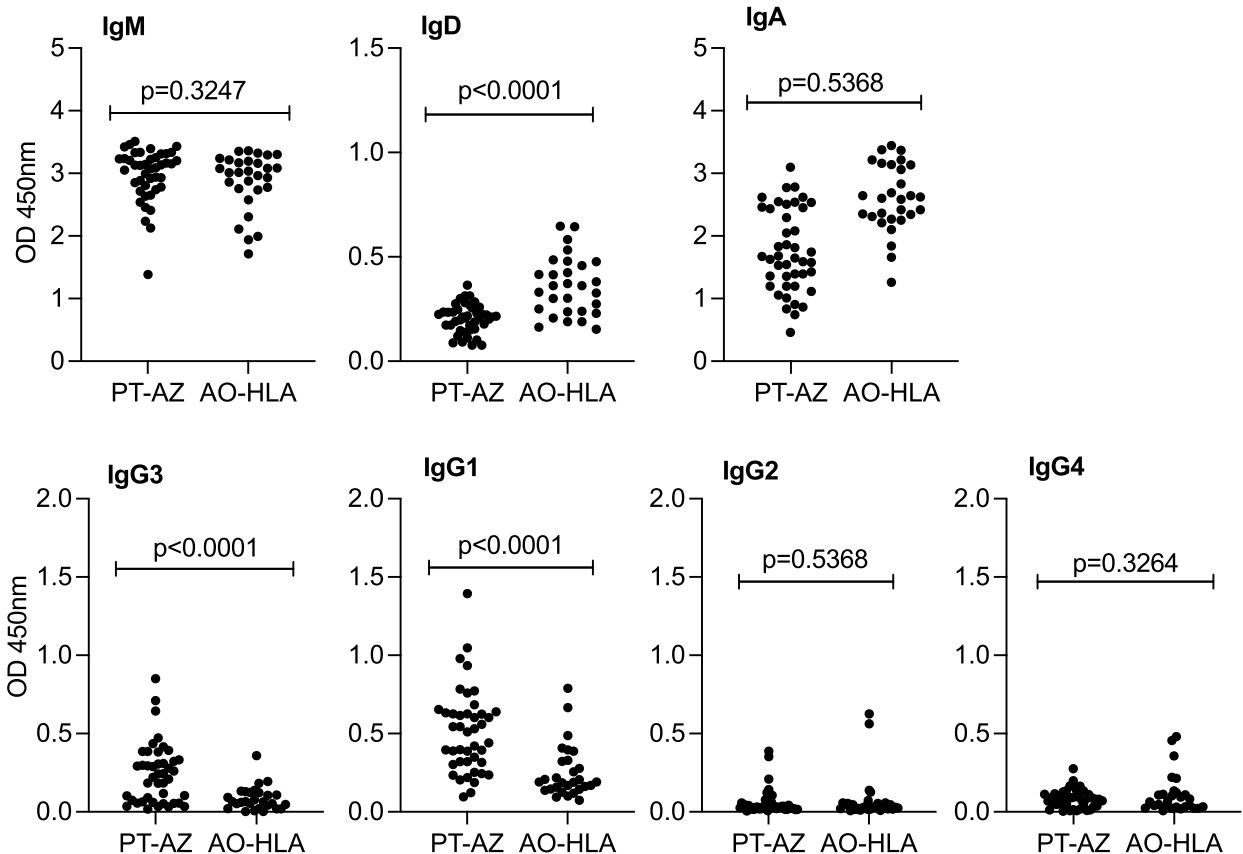

**Fig 6. Analysis of immunoglobulin class M, D and A, and IgG isotypes in MAT+ LiC IgG+ sera panels (AUC>0.995).** Scatter plots of *L. interrogans* specific IgM, IgD, IgA and IgG3, IgG1, IgG2, IgG4 in a serum panel from Portugal (PT-AZ, n = 43) and a panel from Angola (AO-HLA, n = 29). Statistics by Welch's t test.

## Discussion

The current definitive serologic test for leptospirosis is the microscopic agglutination test (MAT) which has been used for 100 years. Infection can be confirmed *if paired* acute and convalescent samples from the same patient show a 4X increase in antibody titer [11]. In addition, MAT allows for identification of the serogroup of the infecting serovar. This is important information to acquire in areas where a medley of pathogenic, intermediate and non-pathogenic *Leptospira* strains circulate. Thus, MAT is the assay of choice to confirm active infection. However, the following major limitations have been noted: MAT cannot be used for immediate case identification because it is insensitive in early infection [11]; it is not suitable for epidemiologic studies [18]; it is unable to distinguish between IgM and IgG [19]; it requires maintenance of large collections of different *Leptospira* serovars that can only be cultured in expensive specialized reference laboratories managed and operated by highly trained personnel. Furthermore, MAT is subjective to human interpretation and cross-reactivity among several *Leptospira* serovars and false positive reactions due to auto-agglutination have also been reported [11,12,20]. Given these weaknesses, several molecular techniques are currently being used for diagnosis of acute infection. These techniques can detect the etiologic agent itself in blood or urine by PCR amplification of *Leptospira* DNA or RNA [21–26]. However, there still

is a need for versatile and affordable serologic assays that can be performed routinely in clinical laboratories in endemic areas where circulating strains are well known.

A recent study showing high incidence of leptospirosis in Vanuatu highlighted the need for improved diagnostic capabilities in developing countries [27]. Interest in redefining the gold standard testing for laboratory diagnosis of leptospirosis has been mounting recently and led to the proposal of several options: combinations of molecular tests and MAT [28,29], as well as molecular tests and EIA [30–32] [25,33–38]. Molecular tests have dramatically improved patient care and reduced deaths due to leptospirosis in a developed country [25]. However, molecular tests do not offer information on the immunity status of the individual. This is important as it has been recently shown that infection of mice with *L. interrogans* (sv Copenhageni, sv Manila, sv Icterohaemorrhagiae) induce an antibody based immune response that protects against homologous re-infection [39]. This opens the field of Leptospirosis diagnostics to applications of easy to deploy serologic assays. We and others [18,29,40] explored the use of a ubiquitous and highly sensitive technique—EIA—to develop an in-house serologic test for laboratory diagnosis of leptospirosis.

We found that in Angola where the predominant circulating *Leptospira* serovars are *L. interrogans* Copenhageni and *L. borgpetersenii* Ballum Arborea a combination of the *L. interrogans* recombinant proteins LigA, LigB and LipL32 correctly identified antigen-specific IgG from patients with clinical and laboratory confirmed leptospirosis (MAT+) with 92% sensitivity and specificity ~ 97% (AUC 0.974) in the province of Luanda and Huambo (AO LDA/HBO), whereas a combination of whole cell extracts of *L. interrogans* sv Copenhageni (LiC), *L. kirschneri* Mozdok (LkM), *L. borgpetersenii* Arborea (LbA) and *L. biflexa* Patoc (LbP) accurately identified patients with clinical and laboratory confirmed leptospirosis (MAT+) with 100% sensitivity and specificity ~ 98% for both provinces (AUC 0.997 for LDA/HBO and 1.000 for HLA). Others have shown that combinations of recombinant proteins containing rLigA, rLigB and rLipL32 on EIA detected antigen-specific IgM and IgG in serum from leptospirosis patients from Peru with 82% sensitivity and 86% specificity compared to MAT [41]. Further, others found that LigA-IgM EIA was more sensitive, but not more specific, than whole-cell based IgM EIA for the early diagnosis of leptospirosis in the Philippines [42]. Another study done recently using a commercial, whole-*Leptospira*-based IgM EIA, reported low accuracy [43]. However, a weakness of that study was the comparison between different classes of tests using a molecular test as the gold standard to gauge sensitivity and specificity of a serologic test. These two different classes of tests complement each other given that a molecular test is expected to detect the infectious agent early in the course of infection, whereas a serologic test should be positive 7–14 days post a new infection. Furthermore, it was recently found that long term production of IgM to *Leptospira interrogans* sv Copenhageni and sv Manilae correlate with colonization of the kidney, in contrast to sv Icteroheamorrhagiae which induced a classical temporary IgM response without kidney colonization [44]. These data further support development of EIA based serologic assays for research purposes.

For Portugal, the best diagnostic performance was achieved with a combination of whole cell extracts of *L. interrogans* sv Copenhageni (LiC), *L. kirschneri* sv Mozdok (LkM), *L. borgpetersenii* sv Arborea (LbA) and *L. biflexa* sv Patoc (LbP) which accurately identified patients with clinical and laboratory confirmed leptospirosis (MAT+) with 100% sensitivity and ~98% specificity (AUC 1.000). In-house ELISAs done by others using whole cell extracts of the most prevalent serovar in Germany (*Leptospira kirschneri* sv Grippotyphosa) performed with a clinical sensitivity of 83% and clinical specificity of 98.5% in MAT+ serum [18], which is equivalent to our results in regards to specificity (98%). However, in our study, whole-cell in-house EIAs made with *L. interrogans* sv Copenhageni produced higher sensitivities in MAT+ serum (93%-98%) than the study in Germany, which could be attributed to the very high prevalence of this

serovar in Portugal. Our data also suggests that sensitivity of the EIA can be improved by adding extracts from other prevalent serovars in the region such as the leptospires isolated from the Angola specimens [10]. Our data further supports [40] the development of in-house IgG EIAs using *Leptospira* serovars known to circulate in the region/country [29] as an alternative method for the diagnosis of Leptospirosis. Overall, the ease of use and accuracy of the EIA make them excellent serologic alternatives to the laborious MAT for diagnostic screening and for epidemiologic studies in resource-limited countries, as we offer an example for Angola.

Interestingly, we found that MAT+ IgG+ serum from Angola had a significantly higher presence of IgD and that IgG3/IgG1 isotypes were significantly increased in the MAT+ IgG + serum from Portugal. As expected, there were no significant differences in IgM or IgA between the two panels given that both sera panels used had equivalent levels of IgG as this was the criteria to do the IgG subtyping analysis. Circulating IgD is found at low levels in serum, it has a short half-life and its function is not clear. Because there is a FcR for IgD in CD3 T cells, it has been proposed that IgD might serve as a bridge for antigen presentation by B cells to T cells [45]. Presence of transient IgD in serum may be indicative of a pre-adaptive immune response ongoing in acute leptospirosis when mature B cells producing IgM+/IgD + reach the spleen. Increased IgD in the Angola panel suggests a very early stage of disease. This is corroborated by the clinical inclusion criteria for patients in the Angola study, which was fever. IgG3 is the first IgG to appear in serum as switching from IgM/D to IgG takes place and it is an early effector, independent of T cell help [46]. IgG1 soon follows after T cell help has been engaged. Presence of IgG3/IgG1 in the Azores panel suggests that these patients also presented at the hospital with acute leptospirosis. Production of IgG2 and IgG4 is associated with long exposure to antigen and switching to IgG4 is associated with induction of tolerance [46,47]. Absence of IgG2 and IgG4, two isotypes commonly associated with later stages of disease, further corroborate the clinical characterization of the patients included as acute leptospirosis. Thus, immunoglobulin G isotyping provides another measure that can aide in clinical characterization and should be further tested to discriminate infection from reinfection.

The increased sensitivity and specificity of this assay may be due to the use of Bugbuster reagent in our protein extraction protocol. This reagent breaks up the *Leptospira* membrane gently but preserves integrity of the proteins. The *Leptospira* LPS induces specific immune responses to each serovar leading to low cross-reactivity/specificity between strains, and thus it defines the serovars. Thus, the high specificity of our assay might be associated to the protein extracted by the Bugbuster reagent. This is further corroborated by the detection of IgG3/IgG1 rather than IgG2/IgG4 given that IgG1 and IgG3 are generally induced by protein antigens, whereas IgG2/IgG4 are generally induced by polysaccharide antigens (in this case, *Leptospira* LPS) [45,46].

One limitation of this study was that we did not have access to serum from healthy individuals from the endemic areas used in this study (Portugal and Angola). Use of serum from healthy individuals from an endemic area may lower the overall specificity of the assay. Another limitation is that serum from other diseases used in differential diagnosis of leptospirosis in co-endemic areas (e.g. malaria) was not available for cross-reactivity testing.

In conclusion, in areas where circulating serovars of pathogenic *Leptospira* are well defined, accurate assays such as EIA which can be operated at a fraction of the cost of MAT should be explored as effective serodiagnostic tools for leptospirosis applicable to humans and animals. Addition of IgG3/IgG1 isotyping further increases likelihood that the samples tested represent acute human leptospirosis.

## Supporting information

**S1 Fig. Purified recombinant proteins rLigA, rLigB and rLipL32.** The SDS-PAGE gel was stained by coomassie blue to confirm the molecular weight of the protein and antigenicity was confirmed by western blot of proteins electrotransferred into PVDF membranes against antigen-specific mouse polyclonal antibodies.
(EPS)

**S2 Fig. Potential sensitivity of the lead recombinant protein candidates.** IgG specific to 5 *L. interrogans* recombinant proteins was determined in serum from 6 C3H-heJ mice infected with $10^7$ *L. interrogans* Copenhageni by ELISA; serum from 6 non-infected controls were tested against LigA to establish the cutoff.
(EPS)

**S1 Table. Diagnostic performance of other *Leptospira* extracts.** LDA/HBO, Luanda/Huambo; HLA, Huila; AZ, Azores, LIS, Lisbon; ROC(AUC), receiver-operating characteristic (ROC) analysis with area under the curve (AUC).
(DOCX)

**S1 Data. Excel file containing all the data used to make the figure plots and tables.**
(XLSX)

**S1 Text. Step-by-step EIA protocol for detection of anti-*Leptospira* antibody in human serum.**
(DOCX)

## Acknowledgments

We thank Dr. David Haake (UCLA VA), Dr. Odir Dellagostin (Univ. Federal de Pelotas, Brazil) and Dr. Jarlath Nally (USDA-ARS) for providing some of the *Leptospira* serovars used in this study. We also thank Alan McBride (Univ. Federal de Pelotas, Brazil) for providing leptospirosis serum used to acquire preliminary data that supported pursuit of this project.

## Author Contributions

**Conceptualization:** Maria Gomes-Solecki.

**Data curation:** Mariana Soares Guedes, Maria Gomes-Solecki.

**Formal analysis:** Elsa Fortes-Gabriel, Mariana Soares Guedes, Advait Shetty.

**Funding acquisition:** Maria Gomes-Solecki.

**Investigation:** Elsa Fortes-Gabriel, Mariana Soares Guedes, Advait Shetty.

**Methodology:** Elsa Fortes-Gabriel, Mariana Soares Guedes, Advait Shetty, Charles Klazer Gomes, Teresa Carreira, Lisa Esteves.

**Project administration:** Maria Gomes-Solecki.

**Resources:** Maria Luísa Vieira, Luísa Mota-Vieira.

**Supervision:** Maria Gomes-Solecki.

**Writing – original draft:** Maria Gomes-Solecki.

**Writing – review & editing:** Maria Gomes-Solecki.

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
