## [Decision Letter · Decision Letter 0]

3 Jan 2022

Dear Dr. Gomes-Solecki,

Thank you very much for submitting your manuscript "Enzyme immunoassays (EIA) for serodiagnosis of human leptospirosis: IgG3/IgG1 can help define acute disease" for consideration at PLOS Neglected Tropical Diseases. As with all papers reviewed by the journal, your manuscript was reviewed by members of the editorial board and by several independent reviewers. The reviewers appreciated the attention to an important topic. Based on the reviews, we are likely to accept this manuscript for publication, providing that you modify the manuscript according to the review recommendations. I apologize for the long review process for this manuscript; it was difficult to recruit reviewers and one who agreed was so late that I have provided only two reviews.

One significant point raised in the reviews is the age of the sera used, and how the collection and storage conditions might affect the results. Please address this point in the text by either performing some tests of your ELISA-based diagnostics with fresh sera or by acknowledgint the possible limitations of using sera collected years ago.

Sincerely,

Jenifer Coburn, PhD

Associate Editor

Richard Phillips

Deputy Editor

Reviewer's Responses to Questions

**Key Review Criteria Required for Acceptance?**

**Methods**

-Are the objectives of the study clearly articulated with a clear testable hypothesis stated?

-Is the study design appropriate to address the stated objectives?

-Is the population clearly described and appropriate for the hypothesis being tested?

-Is the sample size sufficient to ensure adequate power to address the hypothesis being tested?

-Were correct statistical analysis used to support conclusions?

-Are there concerns about ethical or regulatory requirements being met?

Reviewer #1: (No Response)

Reviewer #2: -Are the objectives of the study clearly articulated with a clear testable hypothesis stated? yes

-Is the study design appropriate to address the stated objectives? yes

-Is the population clearly described and appropriate for the hypothesis being tested? yes

-Is the sample size sufficient to ensure adequate power to address the hypothesis being tested? yes

-Were correct statistical analysis used to support conclusions? yes

-Are there concerns about ethical or regulatory requirements being met? no

**Results**

-Does the analysis presented match the analysis plan?

-Are the results clearly and completely presented?

-Are the figures (Tables, Images) of sufficient quality for clarity?

Reviewer #1: (No Response)

Reviewer #2: Does the analysis presented match the analysis plan? yes

-Are the results clearly and completely presented? yes

-Are the figures (Tables, Images) of sufficient quality for clarity? yes

**Conclusions**

-Are the conclusions supported by the data presented?

-Are the limitations of analysis clearly described?

-Do the authors discuss how these data can be helpful to advance our understanding of the topic under study?

-Is public health relevance addressed?

Reviewer #1: (No Response)

Reviewer #2: -Are the conclusions supported by the data presented? yes

-Are the limitations of analysis clearly described? methodology needs more detail, see comments above

-Do the authors discuss how these data can be helpful to advance our understanding of the topic under study? yes

-Is public health relevance addressed? yes

**Editorial and Data Presentation Modifications?**

Reviewer #1: (No Response)

Reviewer #2: Line 245. There is a extra % in the end of the text.

**Summary and General Comments**

Reviewer #1: The paper reported affordable in-house enzyme immunoassays using Leptospira serovars whole protein extracts and recombinant proteins, individually and in combination. The test can be performed for laboratory confirmation of leptospirosis to replace the laborious microscopic agglutination test (MAT). Interestingly, the authors showed increased sensitivities when a combination of antigens were used. This approach has been used previously to increase diagnostic sensitivity in many diseases. In general, the paper addresses an important issue, since the gold standard test (MAT) has several limitations. However, there are some concerns which should be considered. 

• The study lake sera of healthy individual from same endemic areas/countries. These sera are better to assess specificity of the test. 

• Purity of the recombinant proteins and their molecular sizes are not shown in Figures.

• It is not clear why the authors used sera of infected mice.

• In line 30: put the acronym (AO) in full when come first. 

• Fig 1 could be omitted. 

• In line 261: it is stated that whole cell extract from eight Leptospira serovars were used, however, fig 5, table 2 and table S1 show only results of 4 serovars. Some results are probably missing!

• In EIA protocol, please include Ph of the coating buffer (sod carbonate) and content/nature of the washing buffer.

• Line 718: please indicate whether substrate was incubated at dark? 

• For figs 3-6 and S1, it is not clear what do the horizontal lines represent?

Reviewer #2: Fortes-Gabriel et al, describe an Enzyme immunoassays (EIA) for serodiagnosis of human leptospirosis and describe that IgG3/IgG1 can help define acute disease. This is a great effort to improve human serological diagnosis in Leptospirosis as an alternative to the laborious and expensive MAT for screening acute infections in areas where circulating serovars of pathogenic Leptospira are well defined.

Although the article is clearly described, there is room for improvements such as the following:

1. The serum collection is old (2010 and therefore the quality of these sera can be affected by the preservation methods. The text does not indicate whether they were preserved at -20 ° C or -80 ° C with or without glycerol. This maintenance directly affects the quality and preservation of the immunoglobulins, which can give variable results. This should be included in the methodology and discussed in the results.

2. Date of collection of the serum are not mentioned for PT (AZ), PT (LIS) and healthy patients from Florida

3. Line 61-62. Brucellosis is another zoonotic disease common as differential diagnosis in the African continent. Please consider this differential to be include it.

4. It is recommended to include animal reservoirs in the countries that participate and sources of contagion in the description of the disease so the One Health concept is fully addressed.

5. Could this immunoassay be adapted and be useful in animals in these regions? this can be included in the discussion

PLOS authors have the option to publish the peer review history of their article (what does this mean?). If published, this will include your full peer review and any attached files.

Reviewer #1: Yes: Elfadil Abass

Reviewer #2: Yes: Gabriela Hernandez-Mora

Figure Files:

Data Requirements:

Reproducibility:

References

---

## [Editor Report · Decision Letter 1]

9 Feb 2022

Dear Dr Gomes-Solecki,

We are pleased to inform you that your manuscript 'Enzyme immunoassays (EIA) for serodiagnosis of human leptospirosis: specific IgG3/IgG1 isotyping can further inform diagnosis of acute disease' has been provisionally accepted for publication in PLOS Neglected Tropical Diseases.

Best regards,

Jenifer Coburn, PhD

Associate Editor

Richard Phillips

Deputy Editor

---

## [Editor Report · Acceptance letter]

17 Feb 2022

Dear Dr Gomes-Solecki,

We are delighted to inform you that your manuscript, "Enzyme immunoassays (EIA) for serodiagnosis of human leptospirosis: specific IgG3/IgG1 isotyping may further inform diagnosis of acute disease," has been formally accepted for publication in PLOS Neglected Tropical Diseases.

Best regards,

Shaden Kamhawi

co-Editor-in-Chief

Paul Brindley

co-Editor-in-Chief
